# Facility and care provider emergency preparedness for neonatal resuscitation in Kano, Nigeria

**Fatima Usman** [1]*, **Fatimah I. Tsiga-Ahmed**[2], **Mohammed Abdulsalam**[1], **Zubaida L. Farouk**[1], **Binta W. Jibir**[3], **Muktar H. Aliyu**[4]

**1** Department of Pediatrics, Bayero University, Kano & Aminu Kano Teaching Hospital, Kano, Nigeria, **2** Department of Community Medicine, Bayero University, Kano & Aminu Kano Teaching Hospital, Kano Nigeria, **3** Department of Pediatrics, Hasiya Bayero Pediatrics Hospital, Kano, Nigeria, **4** Vanderbilt Institute for Global Health, Vanderbilt University Medical Centre, Nashville, Tennessee, United States of America

* fusman.pae@buk.edu.ng

## Abstract

### Introduction

The knowledge, attitude, and practice of emergency neonatal resuscitation are critical requirements in any facility that offers obstetric and neonatal services. This study aims to conduct a needs assessment survey and obtain individual and facility-level data on expertise and readiness for neonatal resuscitation. We hypothesize that neonatal emergency preparedness among healthcare providers in Kano, Nigeria is associated with the level of knowledge, attitudinal disposition, practice and equipment availability at the facility level.

### Methods

A semi-structured, self-administered questionnaire was administered to a cross-section of health providers directly involved with neonatal care (n = 112) and attending a neonatal resuscitation workshop in Kano state. Information regarding knowledge, attitude, practice and facility preparedness for neonatal resuscitation was obtained. Bloom's cut-off score and a validated basic emergency obstetric and neonatal care assessment tool were adopted to categorize outcomes. Multivariable logistic regression was employed to determine independent predictors of knowledge and practice.

### Results

Almost half (48% and 42% respectively) of the respondents reported average level of self-assessed knowledge and comfort during resuscitation. Only 7% (95% CI:3.2–13.7) and 5% (95% CI:2.0–11.4) of health providers demonstrated good knowledge and practice scores respectively, with an overall facility preparedness of 46%. Respondents' profession as a physician compared to nurses and midwives predicted good knowledge (aOR = 0.08, 95% CI: 0.01–0.69; p = 0.01), but not practice.

**Data Availability Statement:** Our data contain potentially identifying and sensitive participant information and therefore cannot be made publicly accessible without permission from the Ethics

Review Committee of Kano State Ministry of Health. All data requests should be directed to Mr. Nasir Tafida, Secretary, Research Ethics Review Board, Kano State Ministry of Health, Post office road, Kano Nigeria Email: nasirtafida438@gmail.com Phone: +234 (0)8022759125.

**Funding:** The author(s) received no specific funding for this work.

**Competing interests:** The authors have declared that no competing interests exist.

## Conclusion

Healthcare provider's knowledge and practice including facility preparedness for emergency neonatal resuscitation were suboptimal, despite the respondents' relatively high self-assessed attitudinal perception. Physicians demonstrated higher knowledge compared to other health professionals. The low level of respondents' awareness, practice, and facility readiness suggest the current weak state of secondary health systems in Kano.

## Introduction

Each year, approximately 136 million babies are born, with an estimated 10% requiring some form of basic support at birth, and 3–6% needing active resuscitation [1,2]. Over 90% of neonatal deaths occur in low and middle-income countries of Sub-Saharan Africa (SSA) and southeast Asia [2,3], with a quarter of these deaths related to intrapartum events. Nigeria is among the top three contributors to the global burden of infant mortality [3], with a neonatal mortality rate (NMR) that has stagnated between 37 and 41 per 1000 live births over the past three decades. The high rates of neonatal mortality in Nigeria have been linked to a lack of access to quality emergency services, including resuscitation at birth [4,5]. This finding is partly attributable to the paucity of skilled personnel—Nigeria has a workforce of only 476 Pediatricians [6], and a health worker to patient ratio of 20/10,000, which is below the World Health Organization (WHO) recommendation of 23/10,000 [7]. Kano State, the most populated state in Nigeria, has the highest burden of NMR in the country, at 69 per 1000 live births. Evident NMR disparity exists in the State linked to sociodemographic indices and is worst among infants born to young mothers with no education, low wealth index quintile, male gender, high birth order and birth interval of less than 2 years. Infants born in rural areas are 54% more likely to die than their urban counterparts [8]. Reports from 6 African countries showed that among health workers, the knowledge of neonatal resuscitation was poor, ranging between 2%–12% [1]. In addition, only 8%-22% of the surveyed facilities had appropriate resuscitation equipment [1]. It remains unclear whether these findings are the same in Nigeria, a country with similar demographics, the most populous in Africa with a high fertility rate, that was not included in the study.

The intrapartum transition of the new-born requires meticulous proficiency in the immediate and emergent care provided at birth and is key to a favorable neonatal outcome. A recent meta-analysis reported a 37% reduction in intrapartum asphyxia-related deaths with neonatal resuscitation training (NRT) [9]. To avert the projected annual global death of 2.8 million neonates, 80% of which will likely occur in SSA and southeast Asia [3]; and achieve the proposed Sustainable Development Goal target of reducing preventable neonatal deaths to at least 12 deaths per 1,000 live births by 2030, concerted efforts are needed to strengthen and accelerate the availability of emergency NRT [10]. For NRT to be impactful, the cognitive knowledge and attitude towards it, technical know-how and equipment availability to support proficiency are essential. As a first step, obtaining reliable data for evaluating key action areas that need support is necessary.

We hypothesize that neonatal emergency preparedness among healthcare providers in Kano, Nigeria may be associated with the level of knowledge, attitudinal disposition, practice and equipment availability at the facility level. To our knowledge, there is no published research on the status of emergency preparedness for neonatal resuscitation in the most populous state in Africa's most populated country. We, therefore, conducted a needs assessment

survey to obtain individual and facility-level data on the level of expertise and readiness for neonatal resuscitation. Our multi-center findings will help inform policy to delineate areas that need support and enhance service delivery.

## Methodology

### Study site

This survey was conducted in secondary health facilities in Kano, the capital of Kano state in northwest Nigeria (estimated population: 13 million) [11]. The State has 44 local government areas (LGAs), each divided into wards totaling 484. Health services are provided via three tiers of government: primary health care provided by the local government, secondary health care by the state government, and tertiary care supported by the federal government. Private facilities provide different services at various community levels. The state has 1350 primary health facilities and 40 secondary health centers spread across the LGAs. Only 33 secondary health facilities have a maternity unit and conduct deliveries. The total fertility rate in Kano is 6.5 and only 19.2% of deliveries take place in a health facility [12].

### Study design

A descriptive cross-sectional survey was conducted during a two-day NRT workshop in November 2020.

### Sampling

The State Ministry of Health organized a two-day neonatal resuscitation workshop for healthcare providers in secondary health centers of the state. The training was conducted in two batches. A master health facility list was used to identify the centers that offer obstetric and newborn services and invitations were sent to participants through the state ministry of health and management board. Two participants were invited from each of the 33 secondary facilities that conduct deliveries. The remaining participants were drawn from the two Pediatric secondary health centers within Kano metropolis (Hasiya Bayero Pediatric Hospital and Khalifa Isyaka Rabiu Pediatric Hospital). Using each center's work rota, a convenience sample was used for the selection process, inviting only staff that were off duty to eliminate the risk of staff shortage on active duty during the training period. The diversity of the participants including doctors, nurses, and midwives added strength to the sample and reduced selection bias. Only healthcare providers who were directly involved in neonatal care were invited and participation in the survey was voluntary.

### Sample size estimation

With a power of 80%, 95% confidence level, a desired level of precision of 0.05 and 3% prevalence of healthcare workers with good knowledge of neonatal resuscitation obtained from a similar study in North-eastern Nigeria [13], and a non-response rate of 10%, we determined a minimum sample size of 98 healthcare workers. To increase precision, all healthcare workers (112) who were present at the training were included in the study.

### Study population

The study population was comprised of medical officers, nurses and midwives working in the operating room, delivery, newborn, and Pediatric wards of government-owned secondary health centers in Kano, Nigeria. We excluded any healthcare provider who was not directly

involved in labor/delivery and newborn care, and those who did not provide consent, a prerequisite for inclusion.

## Consent

Participants were informed about the survey at the beginning of the workshop before the commencement of the training, and written consent was obtained from each participant before inclusion in the survey.

## Data collection instrument

A literature review of methods and data collection instruments used for NRT needs assessment was conducted. A semi-structured questionnaire was developed from core components of the neonatal resuscitation algorithm [14,15] and standardized questions from similar studies [16–18] pertinent to our practice. The questionnaire was self-administered and completed before the start of the training to ensure efficiency, anonymity and reduce response bias. It comprised a total of 51 questions divided into five sections covering information on sociodemographic/work-related characteristics, knowledge of neonatal resuscitation, practical skills, self-reported readiness, and facility-related preparedness. The final questions used were consensually reviewed, edited, and validated, and were pretested among 12 neonatal health care providers consisting of nurses and doctors in a tertiary institution.

**Outcome variables.** The outcome variables considered were four, namely: knowledge; self-reported preparedness and attitude; practice; and facility preparedness for neonatal resuscitation. The knowledge on neonatal resuscitation was assessed using 12 elements on the questionnaire, eliciting information on the appropriate sequence of resuscitation, initial steps and airway management, indications for ventilation and chest compression, rate of ventilation, indications for oxygen use and targeted oxygen saturation, when to use drugs, and the indicators of adequate resuscitation. Self-reported preparedness and attitude were evaluated using four questions. Two questions elicited self-assessed knowledge and opinion regarding facility preparedness for neonatal resuscitation with a 6-level Likert-type scale. Other questions enquired about the participants' attitude during resuscitation, specifically the need to call for help and how soon the request was made, and their comfort level during resuscitation.

The practice of the respondents was assessed using 15 items on resuscitation technique and everyday practice, e.g., partograph use to predict the need for resuscitation, the number of staff that participate during resuscitation, neonatal temperature regulation, routine umbilical cord care practice, expertise in umbilical catheterization, etc. Facility level preparedness was audited with 8 questions in three domains: availability of essential and priority resuscitation equipment, and on-the-job staff training. The first two domains had questions on available equipment for neonatal resuscitation and the third domain covered the presence of neonatal resuscitation guidelines and equipment checklist.

**Explanatory variables.** Demographic information included age, sex, the profession of the participant (physician, nurse, or midwives), unit/ward of practice, years of experience in the specified ward, number of deliveries conducted, and neonatal resuscitation events performed by the respondents during the preceding year; and whether the respondent had received NRT in the past. Other information captured the availability of advanced equipment and certain emergency medical facilities/services, the total number of deliveries and deaths within 24 hours and the number of staff dedicated to neonatal care.

## Scoring of responses

One point was allocated for each correct response and zero was allocated to a wrong or 'don't know' response. For each section, the mean score was calculated as a percentage of the total for that section (12 points equivalent to 100% for knowledge and 15 points equivalent to 100% for practical skills). Bloom's cut-off [19] of 80% was adopted to categorize knowledge and practice. A score of ≥9.6 was considered good knowledge and a score of ≥12 was considered good practice.

The WHO method of assessing facility service availability and readiness for basic emergency obstetric and neonatal care (BEmONC) [20] was used to evaluate facility-related level of preparedness for neonatal resuscitation. Three domains with 29 indicators were assessed namely: staff and training domain comprising of two indicators; essential equipment and drugs domain consisting of 15 indicators; and priority equipment domain consisting of 12 indicators. A composite score was created for the indicators in the assessed domains, with each item per domain scored as one if available and zero if absent. The scores were tallied to obtain a total score for each domain and converted to a percentage. The expected target was 100%, so each domain was allocated 33.3% of the overall score. The readiness of a facility to provide optimal neonatal resuscitation was calculated by adding the percentage proportions of the three domains. An overall score of 50% [20,21] was considered as the cut-off for a well-equipped and prepared facility.

## Data privacy and quality assessment

Paper forms were used to collect data anonymously, and these were distributed at the beginning of the training before didactic sessions began. All questionnaires were checked for completeness, correctness, clarity, and consistency by the investigators immediately after collection from participants. The completed forms were kept in a secured institutional unit accessible only to other investigators on request.

## Data analysis

All analyses were conducted using STATA 15.0 (STATA Corp, College Station, TX, USA). Proper coding and categorization of data were done, and these were rechecked for completeness and accuracy. Frequencies and percentages were described for categorical variables, mean (with standard deviation, SD) was reported for age and median with interquartile range (IQR) for years of experience. Univariate analysis was conducted to identify variables associated with knowledge and practice of neonatal resuscitation. All variables were considered a priori confounding variables for both knowledge and practice of neonatal resuscitation as identified by previous literature [18,22–25]. Independent variables with $p < 0.10$ at the bivariate level were included in the multivariate analysis. To generate adjusted odds ratios, a forward selection approach to modelling was employed, and a parsimonious model was built with retention of those variables that changed the odds ratio by at least 10%. A p-value $<0.05$ was considered statistically significant.

## Ethics approval

Ethical approval was obtained from the Kano State Ministry of Health research ethics committee (Ref: MOH/Off/797/T.I/2111). Signed consent was obtained from all the respondents and participation was voluntary. All provisions of the Helsinki declaration were respected.

## Results

A total of 112 participants responded to the survey and 111 were included for analysis Fig 1. The age of participants ranged between 20 to 56 years (mean ± SD: 34.6 ± 8.4 years). Approximately half of the respondents (51%, n = 56) work in the delivery room while less than half (46%, n = 50) of the respondents had previous NRT. Approximately 46% (n = 23) received the training in the preceding 3 years. The participants' years of work experience ranged from one year to 32 years (median [IQR]: 5 [2,10] years). Other characteristics are presented in Table 1. Overall, the median participants' self-reported monthly number of deliveries and neonatal deaths within 24 hours of delivery in the facilities were 150 (IQR: 60–250) and 5 (IQR: 2,10) respectively.

### Knowledge and attitude towards neonatal resuscitation

Almost half of the respondents (48%, n = 51) reported an average level of self-assessed knowledge towards neonatal resuscitation, while only 9 (9%) and 3 (3%) believed they had poor and excellent knowledge respectively. However, when knowledge was measured objectively, only 8 participants (7%, 95% CI:3.2–13.7) had good knowledge. Respondents' profession was the only factor identified as a predictor of good knowledge; nurses and midwives were 92% less knowledgeable than physicians (aOR = 0.08, 95% CI: 0.01–0.69; p = 0.01). Age, gender, ward, years of work experience, previous NRT, number of deliveries attended, and neonates resuscitated were not predictive of knowledge Table 2. Comparing the knowledge of neonatal resuscitation between participants from the 33 secondary health centres that conduct deliveries (n = 66, mean score 50.4 ±12.9, 95% CI 47.2–53.6) and participants from the two paediatric hospitals within Kano metropolis (n = 45, mean score 51.1±12.1, 95% CI 47.5–54.8), there was

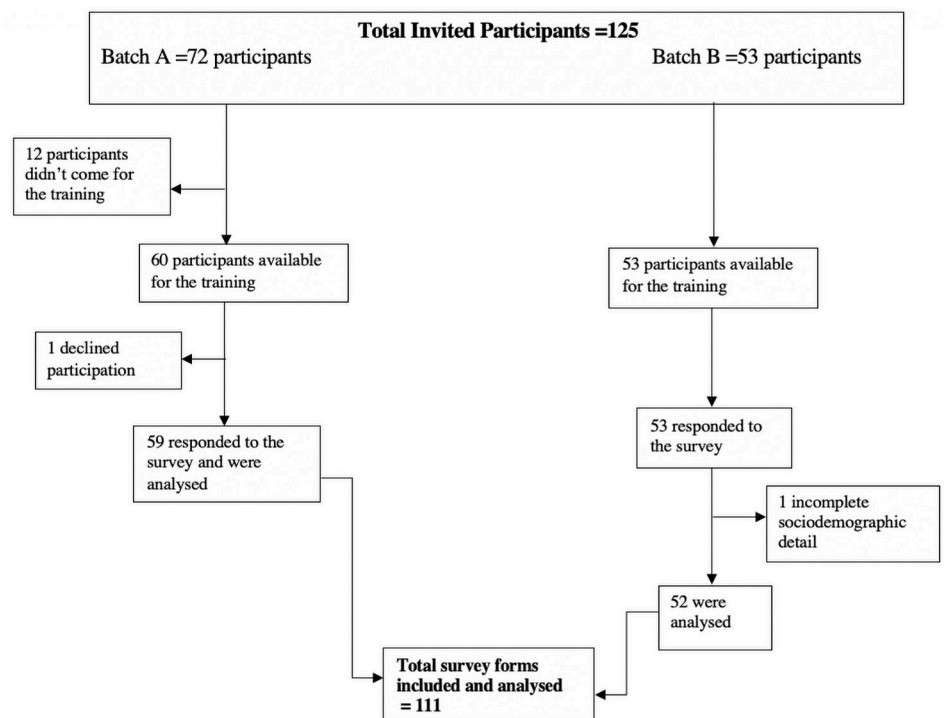

**Fig 1. Schematic representation of the respondents from secondary health facilities in Kano, Nigeria during neonatal resuscitation training.**

**Table 1. Characteristics of the respondents.**

| Variable | Frequency N = 111 | Percent (%) |
|---|---|---|
| **Age in years (*5)** | | |
| 20–29 | 30 | 28.3 |
| 30–39 | 51 | 48.1 |
| 40–49 | 15 | 14.2 |
| 50–59 | 10 | 9.4 |
| **Sex (*0)** | | |
| Male | 45 | 40.5 |
| Female | 66 | 59.5 |
| **Profession (*0)** | | |
| Doctor | 42 | 37.8 |
| Nurse | 44 | 39.6 |
| Midwife | 25 | 22.5 |
| **Ward (*0)** | | |
| Neonatal | 15 | 13.5 |
| Labor/Delivery | 56 | 50.5 |
| Operating Theatre | 9 | 8.1 |
| Pediatrics | 26 | 23.4 |
| Multiple wards | 5 | 4.5 |
| **Work experience in years (*0)** | | |
| 1–5 | 63 | 56.8 |
| More than 5 | 48 | 43.2 |
| **Received NRT (*2)** | | |
| Yes | 50 | 45.9 |
| No | 59 | 54.1 |
| **Deliveries attended in the preceding year (*4)** | | |
| Less than 10 | 29 | 27.1 |
| 10 or more | 78 | 72.9 |
| **Neonates resuscitated in the preceding year (*2)** | | |
| ≤10 | 57 | 52.3 |
| >10 | 52 | 47.7 |

NRT- Neonatal Resuscitation Training.

*—Missing numbers.

no significant difference between the scores (p = 0.76). The majority of respondents (43%, n = 45) self-reported average level of comfort with their newborn resuscitation practice. Furthermore, most respondents (88%, n = 93) often felt the need to call for help during resuscitation, with 75 (77%) making the request as soon as the need is identified, while 10 (10%) ask for assistance only when in the theatre.

## Practice during neonatal resuscitation

Only 5% (95% CI:2.0–11.4) of the respondents (n = 6) had good neonatal resuscitation practices. After adjusting for the effect of previous NRT, no factor was found to be independently associated with good practice Table 2. There was, however, a statistically significant difference (p = 0.03) between the practice scores of participants from the 33 secondary health centers that conduct deliveries (n = 66, mean score 50.6 ±15.0, 95% CI 46.9–54.3) and participants from

**Table 2. Factors associated with knowledge and practice of neonatal resuscitation.**

| | Knowledge | | | | Practice | | | |
|---|---|---|---|---|---|---|---|---|
| Factor | Crude OR* (95% CI) | P-Value | Adjusted OR[a] (95% CI) | P-Value | Crude OR[b] (95% CI) | P-Value | Adjusted OR[c] (95% CI) | P-Value |
| **Age (in years)** | | | | | | | | |
| ≤35 | Reference | | Reference | | Reference | | Reference | |
| >35 | 0.28 (0.03–2.45) | 0.19 | 0.24 (0.03–2.19) | 0.21 | 0.35 (0.04–3.26) | 0.32 | 0.29 (0.03–2.73) | 0.13 |
| **Gender** | | | | | | | | |
| Male | Reference | | | | Reference | | | |
| Female | 0.68 (0.16–2.9) | 0.61 | 0.84 (0.18–3.84) | 0.60 | 0.32 (0.56–1.83) | 0.18 | 0.51 (0.08–1.80) | 0.47 |
| **Profession** | | | | | | | | |
| Doctor | Reference | | Reference | | Reference | | Reference | |
| Nurse/Midwife | 0.07 (0.01–0.62) | 0.002 | 0.08 (0.01–0.69) | 0.01 | 0.28 (0.05–1.62) | 0.14 | 0.46 (0.07–2.84) | 0.10 |
| **Ward** | | | | | | | | |
| Neonatal/Pediatrics | Reference | | | | Reference | | | |
| Labor/Operating theatre | 1.63 (0.30–8.79) | 0.54 | 1.83 (0.03–10.06) | 0.55 | 0.29(0.05–1.68) | 0.15 | 0.35 (0.06–2.09) | 0.25 |
| Multiple | 4.88 (0.36–66.41) | | 4.11 (0.29–57.62|) | | - | | - | |
| **Work experience (in years)** | | | | | | | | |
| 1–5 | Reference | 0.26 | Reference | 0.31 | Reference | 0.15 | Reference | 0.19 |
| >5 | 0.41 (0.08–2.14) | | 0.40 (0.78–2.08) | | 0.25 (0.03–2.19) | | 0.23 (0.03–2.08) | |
| **Deliveries attended to in the preceding year** | | | | | | | | |
| ≤10 | Reference | 0.50 | Reference | 0.57 | Reference | 0.73 | Reference | 0.82 |
| >10 | 0.59 (0.13–2.66) | | 0.63 (0.14–2.85) | | 0.73 (0.13–4.21) | | 1.24 (0.21–7.34) | |
| **Neonates resuscitated in the last year** | | | | | | | | |
| ≤10 | Reference | 0.49 | Reference | 0.50 | Reference | 0.91 | Reference | 0.95 |
| >10 | 0.64 (0.14–2.81) | | 0.60 (0.13–2.71) | | 1.10 (0.21–5.72) | | 1.05 (0.20–5.65) | |
| **Previous NRT** | | | | | | | | |
| Yes | Reference | | Reference | | Reference | 0.09 | Reference | 0.16 |
| No | 0.48 (0.11–2.12) | 0.33 | 0.99 (0.21–4.90) | 0.99 | 0.16 (0.02–1.38) | | 0.20 (0.02–1.90) | |
| **Knowledge of neonatal resuscitation** | | | | | | | | |
| Poor | | | | | Reference | 0.51 | Reference | 0.45 |
| Good | | | | | 2.18 (0.21–22.53) | | 2.67 (0.20–35.15) | |

[a]Adjusted for profession, age and previous NRT.

[b]Adjusted for previous Neonatal Resuscitation Training (NRT).

[c] Adjusted for gender, profession, ward, years of experience and previous NRT.

the two pediatric hospitals (n = 45, mean score 57.0±12.1, 95% CI 52.4–61.6) included in the survey.

## Facility preparedness for neonatal resuscitation

On the self-assessed level of facility readiness, slightly over a third of respondents (38%, n = 40) believed their facilities to be averagely ready, while 28 (28%) felt their facilities were poorly prepared. The overall neonatal resuscitation preparedness index was 45.8%, with each of the three domains assessed having less than 50% average score for the measured indicators. The facility readiness based on a 33.3% proportionate contribution of each domain for staff

and training was 13.2%, for essential equipment and drugs were 17.5% and for priority equipment was 15.1%.

None of the facilities had 100% availability of any indicator (equipment, drugs, and training) of readiness, including advanced equipment and emergency services, Table 3. Only a third

**Table 3. Indicators of readiness to provide neonatal resuscitation in secondary health facilities in Kano (N = 111).**

| Staff and training | Availability (N) | Percent (%) |
|---|---|---|
| Resuscitation guidelines | 52 | 46.8 |
| Equipment checklist | 36 | 32.4 |
| **Essential equipment and drugs** | | |
| Gloves | 90 | 81.1 |
| Normal saline | 88 | 79.3 |
| Ambu bag | 86 | 77.5 |
| Suction bulb | 85 | 76.6 |
| Cord ties | 78 | 70.3 |
| Scissors | 77 | 69.4 |
| 10% DW | 71 | 64.0 |
| Clock | 54 | 48.7 |
| Adrenaline | 45 | 40.5 |
| Nasogastric tubes | 36 | 32.4 |
| Towels/Cloths | 33 | 29.7 |
| Infant warmer | 30 | 27.0 |
| Resuscitation table | 29 | 26.2 |
| Sodium bicarbonate | 13 | 11.7 |
| Polythene bags | 11 | 9.9 |
| **Priority Equipment** | | |
| Stethoscope | 101 | 90.1 |
| Syringes | 88 | 79.3 |
| Oxygen concentrator | 87 | 78.4 |
| Suction device | 81 | 73.0 |
| Pulse oximeter | 68 | 61.3 |
| Glucometer | 54 | 48.7 |
| Incubator | 49 | 44.1 |
| 100% oxygen | 37 | 33.3 |
| Endotracheal tube | 13 | 11.7 |
| Transport Incubator | 10 | 9.0 |
| Laryngoscope | 9 | 8.1 |
| Continuous Positive Airway Pressure Ventilation | 6 | 5.4 |
| **Advanced equipment** | | |
| Ventilator | 15 | 13.5 |
| ECG monitor | 5 | 4.5 |
| Laryngeal mask airway | 3 | 2.7 |
| Blood gas analyzer | 1 | 0.9 |
| **Emergency services** | | |
| Backup generator or Solar | 92 | 82.9 |
| Electricity | 73 | 65.8 |
| Ambulance | 67 | 60.4 |
| Communication system with referral centers | 36 | 32.4 |

N-Total number of respondents.

of the facilities had an equipment checklist. Gloves (81.1%, n = 90) and stethoscopes (90.1%, n = 101) were mentioned as the most available essential and priority equipment, respectively.

## Discussion

We found that the level of knowledge and practice of neonatal resuscitation among healthcare providers in secondary health facilities in Kano to be poor, with only 7% and 5% of respondents demonstrating good knowledge and practice, respectively. Similarly, the overall facility-level preparedness for neonatal resuscitation was inadequate at 18%. These findings imply that the quality of neonatal resuscitation provided at secondary health centers in the state requires significant improvement. Both long-term neurobehavioral and cognitive development of children are associated with the quality of immediate care provided in the first few hours of life. The low utilization of delivery services including lack of antenatal care, delayed hospital presentation during labor, preference for unorthodox obstetric care and home deliveries hinders timely obstetric service demand [4] and worsens neonatal outcome. This may be linked with poverty, lack of insight, and community perceived facility inadequacy in providing quality obstetric services. Failure to improve community health-seeking behaviors and the standard of healthcare practice through continuous retraining and re-evaluation and ensuring availability of essential equipment for optimal service delivery could worsen neonatal and child health indices in the state.

The finding of low level of knowledge and practice among respondents despite almost half (46%) of them having had prior NRT was unexpected. This finding could be due to the lack of regular facility audits to ensure an up-to-date level of comprehension and practice [26] of the healthcare providers. Another reason could be the dearth of frequent standardized formal NRTs, which have been shown to reduce early neonatal death [27], change providers' behavior, level and retention of knowledge and practice [28]. The majority of our respondents (47%) were trained in the preceding 3 years. There is no clear recommendation regarding the number of NRTs per year to maintain practice [1].

The proportion of doctors with good knowledge of resuscitation was higher than other healthcare providers ($p$ = 0.01). This finding likely affected the general quality of service offered, as the majority (62%) of the workforce are nurses and midwives. The prevalence of good knowledge and practice of neonatal resuscitation is similar to reports from Gombe in northern Nigeria [13], probably because of similar demographics, although the majority of the facilities surveyed in Gombe were primary health facilities with a few referral centers. In Ghana, the prevalence of good knowledge was 1.9%, [29] slightly lower than this study; however, in that study, all the respondents were midwives as opposed to this study that included physicians. In contrast, studies from western Nigeria (95%) [22] and Ethiopia (53.8%) [23] with nurse participants only showed much higher prevalence rates than our study. This disparity may be associated with the difference in the survey tool used, with fewer and less technical questions assessed in the two studies.

Although only 7.2% of the respondents had good knowledge, this finding did not affect the overall outcome of practice at 5.4%. As most of the respondents' knowledge is poor, it is unlikely that the small proportion with adequate knowledge will have a meaningful impact on the overall quality of practice. Similarly, the identified paucity of equipment in all the centers may have contributed to the pervasive poor practice, since good practice cannot be achieved without the availability of the necessary tools for service delivery. This finding is similar to reports from other developing countries [13,30]. Western Nigeria [22], however, reported a higher level of practice than our study at 49.7%. The score criterion for adequate knowledge was less than the measure used in the current study.

Overall, facility readiness to offer neonatal resuscitation services including availability of equipment was poor, with a preparedness index of 46%, signifying that approximately 16 of the 35 secondary health centers achieved the minimum of 50% readiness score to offer resuscitation service. This finding is higher than the 29.5% obtained in Tanzania [20], a similar low resource country. The latter study, however, had a larger sample size, including private hospitals and dispensaries, with both obstetric and neonatal service readiness evaluated. Further, in the staff and training domain, availability of trained staff contributed to the better outcome for the latter study, which in the current study was not significantly associated with knowledge (p = 0.99) or practice (p = 0.16), and thus was not included in our domain assessment. Similar to audit reports from southern Nigeria on neonatal resuscitation preparedness and equipment availability [31], the shortage of basic consumables and vital resuscitation equipment from our study is also a major concern. This finding poses a considerable challenge in optimal service delivery, contributing to intervention delays, poor quality of practice and consequently high neonatal mortality [8].

On self-assessed reports of knowledge and facility preparedness, few respondents (9% and 32%, respectively) believed they had a below-average level of knowledge and preparedness. Ninety percent of the participants, however, reported above-average comfort level during resuscitation. This finding, compared with the participants' low scores on the objective assessment of knowledge and practice of NRT confirms that most of the healthcare providers lack insight into their level of individual and facility preparedness for neonatal resuscitation. This is a significant barrier to self-motivated learning, improvement, and capacity building.

A strength of this study is that it was a multicenter survey of a sample of respondents from all the secondary health centers in the state. This design provides a diverse representation from all the centers to permit inferences as to the current state of neonatal emergency service delivery in Kano, the largest city in northern Nigeria. In addition, all cadres of health providers directly involved with newborn resuscitation participated, thus improving the generalizability of the data and allowing targeted interventions to improve neonatal care and reduce mortality.

A limitation of the study is the lack of inclusion of primary and tertiary health centers in the state. The resources required to do this were not available and limiting the design to secondary health centers only would not significantly affect the results, as many deliveries take place there due to referrals from primary health centers, which are not equipped to offer such services. Direct observation of respondent's practice during resuscitation would have enhanced the accuracy and objectivity of the results. However, with the correspondingly low level of knowledge and practice obtained from the survey responses, it is unlikely that the findings would have been considerably different.

## Conclusion

Our finding of an alarming substandard level of knowledge and practice in neonatal preparedness among healthcare providers in secondary facilities in Kano, Nigeria indicates weakness of the existing emergency neonatal preparedness system. Doctors showed relatively higher knowledge of neonatal resuscitation compared to other health professionals. This underscores the urgent need for intensified training and retraining of health care providers. We also recommend the provision of essential and priority equipment in all facilities to curtail the lack of facility preparedness and improve neonatal outcomes.

## Supporting information

**S1 File. Data collection instrument.**
(PDF)

## Author Contributions

**Conceptualization:** Fatima Usman, Mohammed Abdulsalam.

**Data curation:** Fatima Usman.

**Formal analysis:** Fatima Usman, Fatimah I. Tsiga-Ahmed.

**Methodology:** Fatimah I. Tsiga-Ahmed.

**Project administration:** Binta W. Jibir.

**Resources:** Binta W. Jibir.

**Software:** Fatimah I. Tsiga-Ahmed.

**Supervision:** Zubaida L. Farouk, Muktar H. Aliyu.

**Validation:** Fatimah I. Tsiga-Ahmed, Mohammed Abdulsalam, Zubaida L. Farouk.

**Writing – original draft:** Fatima Usman.

**Writing – review & editing:** Fatima Usman, Fatimah I. Tsiga-Ahmed, Mohammed Abdulsalam, Zubaida L. Farouk, Binta W. Jibir, Muktar H. Aliyu.

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
