## [Decision Letter · Decision Letter 0]

20 Aug 2021

PONE-D-21-11467

Facility and care provider emergency preparedness for neonatal resuscitation in Kano, Nigeria.

PLOS ONE

Dear Dr. Usman,

Thank you for submitting your manuscript to PLOS ONE. After careful consideration, we feel that it has merit but does not fully meet PLOS ONE’s publication criteria as it currently stands. Therefore, we invite you to submit a revised version of the manuscript that addresses the points raised during the review process. 

Besides the highly important reviewers' comments, we have the following additional remarks:

Authors are encouraged to further clarify outcome variables and provide the questionnaire used in this study as supplementary file. Authors should explain whether this questionnarre was anonymous or not?  Authors should provide unidentified raw data of the study. PLOS journals require that authors make all data underlying the findings described in their manuscript fully available, with exception when ethical or legal restrictions prohibit public sharing of data. Authors may upload their data to a public repository, or include their data as a Supporting Information file. If data cannot be made public, all restrictions and information necessary for submitting data requests should be outlined in the Data Availability statement, which will be published at the top of the article. Authors are encouraged to include and discuss the folloowing relevant and recent publication "Briggs, D. C., & Eneh, A. U. (2020). Preparedness of primary health care workers and audit of primary health centres for newborn resuscitation in Port Harcourt, Rivers State, Southern Nigeria. The Pan African Medical Journal, 36, 68. " ext-link-type="uri" xlink:type="simple">https://doi.org/10.11604/pamj.2020.36.68.22164".

If applicable, we recommend that you deposit your laboratory protocols in protocols.io to enhance the reproducibility of your results. Protocols.io assigns your protocol its own identifier (DOI) so that it can be cited independently in the future. For instructions see: http://journals.plos.org/plosone/s/submission-guidelines#loc-laboratory-protocols. Additionally, PLOS ONE offers an option for publishing peer-reviewed Lab Protocol articles, which describe protocols hosted on protocols.io. Read more information on sharing protocols at https://plos.org/protocols?utm_medium=editorial-emailutm_source=authorlettersutm_campaign=protocols.

We look forward to receiving your revised manuscript.

Kind regards,

Elsayed Abdelkreem, MD, PhD

Academic Editor

PLOS ONE

Journal Requirements:

2. Please include additional information regarding the survey or questionnaire used in the study and ensure that you have provided sufficient details that others could replicate the analyses. For instance, if you developed a questionnaire as part of this study and it is not under a copyright more restrictive than CC-BY, please include a copy, in both the original language and English, as Supporting Information.  If the original language is written in non-Latin characters, for example Amharic, Chinese, or Korean, please use a file format that ensures these characters are visible.

3. Please state whether you validated the questionnaire prior to testing on study participants. Please provide details regarding the validation group within the methods section.

Reviewers' comments:

Reviewer's Responses to Questions

**Comments to the Author**

1. Is the manuscript technically sound, and do the data support the conclusions?

Reviewer #1: Partly

Reviewer #2: Yes

Reviewer #3: Yes

2. Has the statistical analysis been performed appropriately and rigorously? 

Reviewer #1: No

Reviewer #2: Yes

Reviewer #3: No

3. Have the authors made all data underlying the findings in their manuscript fully available?

Reviewer #1: Yes

Reviewer #2: Yes

Reviewer #3: Yes

4. Is the manuscript presented in an intelligible fashion and written in standard English?

Reviewer #1: No

Reviewer #2: Yes

Reviewer #3: Yes

5. Review Comments to the Author

Reviewer #1: • Facility and care provider emergency preparedness for neonatal resuscitation in Kano, Nigeria.

• Abstract

• Abstract should be written in sections of introduction, method, result and conclusion.

• Under abstract it better to define first what mean by emergency preparedness for neonatal resuscitation rather than neonatal resuscitation.

• On line 23, why you proposed an hypothesis “We hypothesize that the high prevalence of asphyxia-related morbidity…”? While your title was preparedness for neonatal resuscitation? This is the same comment for introduction section.

• On line 33, include the confidence interval for the result of Healthcare provider’s knowledge, skills and facility preparedness.

• On line 37, what do you mean by weak state and what is your base line to say this?

• On line 55 to 57, you stated that “Even among health workers, the knowledge of neonatal resuscitation is poor, ranging between 2%–12%.[1] In addition, only 8%-22% of surveyed facilities have appropriate resuscitation equipment.[1]”, if these are known, why you study again?

• On line 77, you stated as your study will help for policy maker. Do you think that a single site cross sectional study can be influence policy makers?

• On line 105 to 109, why you used prevalence of knowledge only for sample calculation, while there are also practices and facility preparedness? Again your sample is different with abstract section (111 Vs 98 and 112 under result section), why this difference?

• On lone 108, you said that, “To increase precision, all healthcare workers who were

• 109 present at the training were included in the study.”, so why sample calculation is necessary in this case if you were collect data from all them?

• On line 114, you excluded those who did not provide consent. Do you think that unable to give consent is exclusion criteria? If yes, who were none respondents?

• On line 117, you used a semi-structured, self-administered questionnaire. Do you think that it is appropriate method self-administrated questionnaire for knowledge and practice? Why not you used interview administered?

• You are using skill, practice and proficiency exchangeable. Therefore, try to use one of it throughout the document.

• Under your analysis section include the p-value you used to select variable for multivariate analysis.

• Under table 1, check male sex is 405%?

• On line 209, you included attitude. But, there were not stated in the introduction or in method section about attitude. Why this happened/from where attitude came? Even no under operational definition and under measurement.

• On line 227, you said that level of practices…, is it a level? If yes, how much level is there?

• In discussion section, try to limit your discussion to your main objectives only.

• Try to copy edit for punctuation (most of your full stop is before your citation bracket in throughout the document).

Reviewer #2: The manuscript presents an analysis of provider and facility readiness for neonatal resuscitation in Kano state, Nigeria. The issue examined is very pertinent to address the variable and often slow progress toward global neonatal mortality targets. Limitations in the design of the research restrict its validity and generalizability, though the findings are illustrative of barriers encountered widely.

Abstract: The abstract accurately describes the study. The degree of precision in the percentages reported for adequate knowledge, practice, and facility preparedness could be reduced to whole numbers.

Introduction: It is very pertinent to highlight that Kano State has the highest NMR and thus the highest burden, as the most populous state. It might be useful to characterize the area further (urban/rural, geographic or political/social challenges).

Reference 9 presents a Delphi review that estimated the mortality reduction with neonatal resuscitation to be around 30%. More recent systematic reviews and meta-analyses provide information from actual trials (e.g. Dol J 2018).

While the hypothesis that poor neonatal emergency preparedness contributes to the high prevalence of asphyxia-specific mortality is valid, later discussion of the very low rate of in-facility delivery in the state reveals another major factor contributing to the burden. Addressing this aspect would give a more complete picture of the needs.

Methods: In the description of participants, it should be made clear that this was a convenience sample. Did the selection criteria for attending the workshop skew the participants toward those with low knowledge and skills, or perhaps did it skew toward more experienced clinical leadership?

In the sample size estimation, it is not clear what difference (of 10%) was being measured.

The data collection instrument regarding essential equipment and drugs inquired about several medications that are no longer recommended by ILCOR (International Liaison Committee on Resuscitation) for acute neonatal resuscitation (calcium, bicarbonate, naloxone).

The reliability of data on total number of deliveries and deaths within 24 hours and staff numbers would seem low if based on report from non-supervisory personnel.

What was the basis for considering a facility readiness score of 50% as sufficient to qualify a facility as well-equipped and prepared?

Results: The percentages of respondents could be rounded to whole numbers.

Discussion: When comparing the results to findings of other studies, there is frequent speculation on cause of differences – e.g. “In Ghana….mainly because all respondents were midwives…”. The demographic differences can be highlighted, but causation should not be implied. On page 19, “Western Nigeria reported…..probably because the score criterion….” and “….in Tanzania….probably because the latter study….” should be similarly revised.

The first paragraph of page 20 regarding strengths of the study over-estimates the strength of the design, as it was not a representative sample, nor was self-report validated by observation or triangulation. This paragraph should be revised or omitted.

The role of low utilization of health facilities for delivery should also be highlighted in the discussion. This may be linked to the inadequate knowledge, skills, and equipment in the facilities.

Conclusion: The conclusion appropriately highlights the value of information gained in the study.

Figures and Tables: Figure 2 adds little to the results as presented in the text, and it could be omitted.

References: References are current and complete.

Reviewer #3: PLOS ONE REVIEW COMMENTS

Many thanks for the opportunity to review this manuscript addressing an important topic that is key to improving newborn outcomes in a developing country. Neonatal resuscitation is key to the achievement of maternal and newborn health SDG targets on neonatal mortality globally. Overall, the manuscript is well written, paying attention to the existing literature and relating to the current findings. A few specifics comments to be addressed or clarified are as below

Abstract – nicely written and succinct

Introduction

- Line 47 “……one-fourth of these deaths…..” revise to …..a quarter of these deaths ….

- Line 55 – 57 “Even among health workers, the knowledge of neonatal resuscitation is poor, ranging between 2%–12%.[1] In addition, only 8%-22% of surveyed facilities have appropriate resuscitation equipment”…….please clarify whether these are global or Nigerian statistics

Methods

Study site

- Private hospitals were not included as part of the study. Does this mean they do not conduct any births since they were not included in the study? Refer to this related survey about Nigerian health care system at https://www.shopsplusproject.org/sites/default/files/resources/SHOPS%20Nigeria%20Private%20Sector%20Health%20Census_6.15.2014%20FINAL.pdf

- Any information on the proportion of births conducted in the three levels of health facilities would be useful to provide the context

Data collection

- How were study participants recruited/informed about the study?

- When and how were the study participants consented to participate in the study?

- When was the structured interview questionnaire administered?

- Who collected the data/administered the structured questionnaire?

Results

- Line 193: Age of participants is better presented by median given the age ranges and distribution

- Line 194 “The majority of the respondents (50.5%, n = 56)…..” this is just half the population and not the majority

- It would be useful to know if there were any differences in knowledge/skills scores between participants (66 from secondary hospitals + 45 from paediatric hospitals)

- Table 3: Bicarbonate….write the chemical name in full e.g. sodium bicarbonate

Discussion

- Well written and relates to other studies in the field to explain any similarities, variations and meanings of the findings

- Page 19, line 307 – 312: “On self-assessed reports of knowledge and facility preparedness, few respondents (8.5% and 32.4%) believed they had a below-average level of knowledge and preparedness, with 90% of the respondents reporting an above-average comfort level during resuscitation. This confirms that the majority of the healthcare providers lack insight into their deficient preparedness both at the individual and facility level, which is a significant barrier for self-motivated learning, improvement and capacity building.” This statement is confusing and unclear. Authors to review and clarify for ease of understanding

6. PLOS authors have the option to publish the peer review history of their article (what does this mean?). If published, this will include your full peer review and any attached files.

Reviewer #1: **Yes: **Yitagesu Sintayehu

Reviewer #2: No

Reviewer #3: **Yes: **DUNCAN N SHIKUKU

---

## [Author Response · Author response to Decision Letter 0]

4 Oct 2021

Responses to reviewers’ comments

Thank you for considering our manuscript for publication in PLOS One. We have addressed the concerns raised by the reviewers in the table below.

Reviewer #1 comments 

Abstract should be written in sections of introduction, method, result and conclusion. 

Response: Done

Under abstract it better to define first what mean by emergency preparedness for neonatal resuscitation rather than neonatal resuscitation 

Response: Done. The sentence has been modified to fit the context.

On line 23, why you proposed an hypothesis “We hypothesize that the high prevalence of asphyxia-related morbidity…”? While your title was preparedness for neonatal resuscitation? This is the same comment for introduction section. 

Response: This has been changed to “We hypothesize that poor neonatal emergency preparedness among healthcare providers in Kano, Nigeria may be associated with the dearth of knowledge, poor attitudinal disposition, lack of good practice and equipment shortage at the facility level.”

On line 33, include the confidence interval for the result of healthcare provider’s knowledge, skills and facility preparedness 

Response: Confidence intervals added.

On line 37, what do you mean by weak state and what is your base line to say this? 

Response: The basis for this conclusion is from the low level of knowledge, skills and facility preparedness found in our study, which were below the pre-specified reference cut-off values used.

On line 55 to 57, you stated that “Even among health workers, the knowledge of neonatal resuscitation is poor, ranging between 2%–12%.[1] In addition, only 8%-22% of surveyed facilities have appropriate resuscitation equipment.[1]”, if these are known, why you study again?

Response:The cited study was done in 6 African countries (Egypt, Ghana, Kenya, Rwanda, Tanzania, and Uganda). It is unclear if findings will be similar in Nigeria, a country with similar demographics that was not included in the cited study.

On line 77, you stated as your study will help for policy maker. Do you think that a single site cross sectional study can be influence policy makers?

Response:The study participants were drawn from all the secondary health centres in the state, hence can be described as a multi-centre survey with the potential to inform policy decisions. 

On line 105 to 109, why you used prevalence of knowledge only for sample calculation, while there are also practices and facility preparedness? Again your sample is different with abstract section (111 Vs 98 and 112 under result section), why this difference?

Response: Two other sample sizes were computed using prevalence of healthcare workers with good resuscitation practices and prevalence of facility preparedness from a similar setting, however, a lower sample size was obtained in both cases. Therefore, the power analysis for prevalence of knowledge was used for this study. 

The sample size has been corrected in the abstract to 112. One participant had incomplete records and was excluded from the final analysis, hence, 111 were finally included. In the methods, 98 was the minimum sample calculated. However, all participants that provided consent were included in analysis, which will improve the precision of the findings

On line 108, you said that “To increase precision, all healthcare workers who were 109 present at the training were included in the study.”, so why sample calculation is necessary in this case if you were collect data from all them?

Response: Sample calculation was necessary to obtain the minimum sample required to avoid a type II error and for objective extrapolation of results. Including all the 112 survey participants may improve the precision of results and is unlikely to impact negatively on the findings.

On line 114, you excluded those who did not provide consent. Do you think that unable to give consent is exclusion criteria? If yes, who were none respondents?

Response: Voluntary participation is an ethical pre-requisite for inclusion. Only one participant declined to take part in the survey.

On line 117, you used a semi-structured, self-administered questionnaire. Do you think that it is appropriate method self-administrated questionnaire for knowledge and practice? Why not you used interview administered?

Response: The questionnaire was self-administered to ensure anonymity and reduce response bias. Further, all the participants were literate and could respond to the survey appropriately based on instructions provided. The interviewer to participants ratio was also low, making interviewer administration less efficient and timely.

You are using skill, practice and proficiency exchangeable. Therefore, try to use one of it throughout the document 

Response:Thank you. Done.

Under your analysis section include the p-value you used to select variable for multivariate analysis

Response: P-value used to include independent variables in the multivariate analysis now included

Under table 1, check male sex is 405%?

Response: We apologise for the typo. The correct number is 40.5% (now corrected).

On line 209, you included attitude. But, there were not stated in the introduction or in method section about attitude. Why this happened/from where attitude came? Even no under operational definition and under measurement.

Response: This information has been added to the abstract, introduction and methods section (outcome variables) to reflect the findings.

On line 227, you said that level of practices…, is it a level? If yes, how much level is there?

Response:This phrase has been modified to practice during neonatal resuscitation.

In discussion section, try to limit your discussion to your main objectives only.

Response: Thank you. Done

Try to copy edit for punctuation (most of your full stop is before your citation bracket in throughout the document).

Response: Thank you. Done

Reviewer #2 comments 

The abstract accurately describes the study. The degree of precision in the percentages reported for adequate knowledge, practice, and facility preparedness could be reduced to whole numbers.

Response:Done

Introduction: It is very pertinent to highlight that Kano State has the highest NMR and thus the highest burden, as the most populous state. It might be useful to characterize the area further (urban/rural, geographic or political/social challenges).

Response:More detail has been added characterizing neonatal mortality in the state based on urban/rural, geographic, or political/social challenges.

Reference 9 presents a Delphi review that estimated the mortality reduction with neonatal resuscitation to be around 30%. More recent systematic reviews and meta-analyses provide information from actual trials (e.g. Dol J 2018).

Response:This has been changed to a more recent systematic review. (https://doi.org/10.1136/bmjpo-2017-000183)

While the hypothesis that poor neonatal emergency preparedness contributes to the high prevalence of asphyxia-specific mortality is valid, later discussion of the very low rate of in-facility delivery in the state reveals another major factor contributing to the burden. Addressing this aspect would give a more complete picture of the needs.

Response:Done, see lines 468-484 on the recised manuscript with track changes

Methods: In the description of participants, it should be made clear that this was a convenience sample. Did the selection criteria for attending the workshop skew the participants toward those with low knowledge and skills, or perhaps did it skew toward more experienced clinical leadership? Response:Convenience sample has been stated as the method of sampling.“Using each centre’s work rota, a convenience sample was used for the selection process, inviting only staff that were off duty to eliminate the risk of staff shortage on active duty during the training period. Response:The diversity of the participants including doctors, nurses, and midwives added strength to the sample and reduced selection bias”

In the sample size estimation, it is not clear what difference (of 10%) was being measured.

Response: Sample size parameters corrected. “difference (of 10%)” was a typographical error

The data collection instrument regarding essential equipment and drugs inquired about several medications that are no longer recommended by ILCOR (International Liaison Committee on Resuscitation) for acute neonatal resuscitation (calcium, bicarbonate, naloxone).

Response:Calcium and naloxone have been removed. Sodium bicarbonate, however, although discouraged during brief CPR may be useful during prolonged arrest after adequate ventilation is established and there is no response to other therapies. Information regarding its availability in the surveyed facilities to assess level of preparedness may still be useful (DOI: 10.1111/apa.15754 )

The reliability of data on total number of deliveries and deaths within 24 hours and staff numbers would seem low if based on report from non-supervisory personnel.

Response: We agree, but this is a survey looking at participants’ self-reported knowledge, among other things. The findings therefore reflect the perception of the respondents. In the results section, this detail has been revised to reflect self-reported monthly deaths and deliveries

What was the basis for considering a facility readiness score of 50% as sufficient to qualify a facility as well-equipped and prepared?

Response:The score was determined using the WHO approach, and facility preparedness indicators identified according to the WHO Service Availability and Readiness Assessment (SARA) Manual (https://apps.who.int/iris/bitstream/handle/10665/149025/WHO_HIS_HSI_2014.5_eng.pdf?sequence=1isAllowed=y)

Results: The percentages of respondents could be rounded to whole numbers.

Response:Done

Discussion: When comparing the results to findings of other studies, there is frequent speculation on cause of differences – e.g. “In Ghana….mainly because all respondents were midwives…”. The demographic differences can be highlighted, but causation should not be implied. On page 19, “Western Nigeria reported…..probably because the score criterion….” and “….in Tanzania….probably because the latter study….” should be similarly revised. 

Response:Done

The first paragraph of page 20 regarding strengths of the study over-estimates the strength of the design, as it was not a representative sample, nor was self-report validated by observation or triangulation. This paragraph should be revised or omitted.

Response: The paragraph has been revised.

The role of low utilization of health facilities for delivery should also be highlighted in the discussion. This may be linked to the inadequate knowledge, skills, and equipment in the facilities 

Response:Done, see lines 468-484 or the revised manuscript with track changes

.

Figures and Tables: Figure 2 adds little to the results as presented in the text, and it could be omitted. Response:Figure 2 has now been removed

Reviewer #3 

- Line 47 “……one-fourth of these deaths…..” revise to …..a quarter of these deaths ….

Response:Done

- Line 55 – 57 “Even among health workers, the knowledge of neonatal resuscitation is poor, ranging between 2%–12%.[1] In addition, only 8%-22% of surveyed facilities have appropriate resuscitation equipment”…….please clarify whether these are global or Nigerian statistics 

Response:This has been modified to “Reports from 6 African countries showed that even among health workers, the knowledge of neonatal resuscitation was poor, ranging between 2%–12%.[1] In addition, only 8%-22% of the surveyed facilities had appropriate resuscitation equipment.[1] It still remains unclear whether these findings are the same in Nigeria, a country with similar demographics, the most populous in Africa with a high fertility rate, that was not included in the study.

Private hospitals were not included as part of the study. Does this mean they do not conduct any births since they were not included in the study? Refer to this related survey about Nigerian health care system athttps://www.shopsplusproject.org/sites/default/files/resources/SHOPS%20Nigeria%20Private%20Sector%20Health%20Census_6.15.2014%20FINAL.pdf

Response:The focus of our survey were Kano state government-owned secondary health care centres, hence the lack of inclusion of private facilities.

Any information on the proportion of births conducted in the three levels of health facilities would be useful to provide the context

Response:Unfortunately, this information is not available.

How were study participants recruited/informed about the study? 

Response:Invitations were sent to participants through the Kano State Ministry of Health Management Board.

When and how were the study participants consented to participate in the study?

Response:Participants were informed about the survey at the beginning of the workshop before commencement of the training Written consent was obtained from each participant before inclusion in the study.

When was the structured interview questionnaire administered? 

Response:The questionnaire was self-administered and completed before the start of the training to ensure efficiency, anonymity and reduce response bias

Who collected the data/administered the structured questionnaire?

Response:The questionnaire was self-administered and completed before the start of the training to ensure efficiency, anonymity and reduce response bias.

Line 193: Age of participants is better presented by median given the age ranges and distribution

Response:The data for age of the participants was normally distributed, hence our choice of mean as measure of reporting.

Line 194 “The majority of the respondents (50.5%, n = 56)…..” this is just half the population and not the majority

Response:This has been changed to “approximately half of the respondents….”

It would be useful to know if there were any differences in knowledge/skills scores between participants (66 from secondary hospitals + 45 from paediatric hospitals) 

Response:Differences in knowledge and skills scores by type of hospital added.

Table 3: Bicarbonate….write the chemical name in full e.g. sodium bicarbonate 

Response:Done

Page 19, line 307 – 312: “On self-assessed reports of knowledge and facility preparedness, few respondents (8.5% and 32.4%) believed they had a below-average level of knowledge and preparedness, with 90% of the respondents reporting an above-average comfort level during resuscitation. This confirms that the majority of the healthcare providers lack insight into their deficient preparedness both at the individual and facility level, which is a significant barrier for self-motivated learning, improvement and capacity building.” This statement is confusing and unclear. Authors to review and clarify for ease of understanding 

Response:This has been reviewed to “Few respondents (9% and 32%, respectively) believed they had a below-average level of knowledge and preparedness. Ninety percent of the participants, however, reported above-average comfort level during resuscitation. This finding, compared with the low participants’ scores on objective assessment of knowledge and practice of NRT confirms that most of the healthcare providers lack insight into their level of individual and facility preparedness for neonatal resuscitation. This is a significant barrier for self-motivated learning, improvement, and capacity building”

Additional remarks 

Authors are encouraged to further clarify outcome variables and provide the questionnaire used in this study as supplementary file.

Response:Done

Authors should explain whether this questionnaire was anonymous or not? 

Response:The questionnaire was self-administered and completed anonymously (Lines 243-244 and 328 of the revised manuscript with track changes)

Authors should provide unidentified raw data of the study. PLOS journals require that authors make all data underlying the findings described in their manuscript fully available, with exception when ethical or legal restrictions prohibit public sharing of data. Authors may upload their data to a public repository or include their data as a Supporting Information file. If data cannot be made public, all restrictions and information necessary for submitting data requests should be outlined in the Data Availability statement, which will be published at the top of the article. 

Response:Our data contain potentially identifying and sensitive participant information and therefore cannot be made publicly accessible without permission from the Ethics Review Committee of Kano State Ministry of Health. All data requests should be directed to Mr. Nasir Tafida, Secretary, Research Ethics Review Board, Kano State Ministry of Health, Post office road, Kano Nigeria

Email: nasirtafida438@gmail.com

Phone: +234 (0)8022759125 

Authors are encouraged to include and discuss the following relevant and recent publication "Briggs, D. C., Eneh, A. U. (2020). Preparedness of primary health care workers and audit of primary health centres for newborn resuscitation in Port Harcourt, Rivers State, Southern Nigeria. The Pan African Medical Journal, 36, 68. https://doi.org/10.11604/pamj.2020.36.68.22164".

Response:Done, see line 543-545

Please state whether you validated the questionnaire prior to testing on study participants. Please provide details regarding the validation group within the methods section.

Response:Lines 257-259: The final questions used were consensually reviewed, edited, and validated, and were pretested among 12 neonatal health care providers consisting of nurses and doctors in a tertiary institution.

In your Data Availability statement, you have not specified where the minimal data set underlying the results described in your manuscript can be found. PLOS defines a study's minimal data set as the underlying data used to reach the conclusions drawn in the manuscript and any additional data required to replicate the reported study findings in their entirety. All PLOS journals require that the minimal data set be made fully available. For more information about our data policy, please see http://journals.plos.org/plosone/s/data-availability.

Response:Our data contain potentially identifying and sensitive participant information and therefore cannot be made publicly accessible without permission from the Ethics Review Committee of Kano State Ministry of Health. All data requests should be directed to Mr. Nasir Tafida, Secretary, Research Ethics Review Board, Kano State Ministry of Health, Post office road, Kano Nigeria

Email: nasirtafida438@gmail.com

Phone: +234 (0)8022759125

---

## [Decision Letter · Decision Letter 1]

29 Nov 2021

PONE-D-21-11467R1Facility and care provider emergency preparedness for neonatal resuscitation in Kano, Nigeria.PLOS ONE

Dear Dr. Usman,

Thank you for submitting your manuscript to PLOS ONE. After careful consideration, we feel that it has merit but does not fully meet PLOS ONE’s publication criteria as it currently stands. Therefore, we invite you to submit a revised version of the manuscript that addresses the points raised during the review process. Authors have adequately addressed most reviewers' and editorial comments. Just minor comments remain.In the conclusion: “The low level of respondents’ awareness, practice, and facility readiness demonstrates the current weak state of secondary health systems in Kano”. Conclusion has to be based on study findings. Since this study didn’t investigate the “state of secondary health systems in Kano”, authors are encouraged to revise this sentence (reducing its strength/certainty) (e.g., replacing demonstrates by suggests or indicates).In the abstract: “Respondents’ designation predicted good knowledge (aOR=0.08, 95% CI: 0.01–0.69; p= 0.01), but not practice”. Authors should better define/specify “designation” (e.g., physicians versus nurses/midwives).Please submit your revised manuscript by Jan 13 2022 11:59PM. If you will need more time than this to complete your revisions, please reply to this message or contact the journal office at plosone@plos.org. Please include the following items when submitting your revised manuscript:A rebuttal letter that responds to each point raised by the academic editor and reviewer(s). You should upload this letter as a separate file labeled 'Response to Reviewers'.A marked-up copy of your manuscript that highlights changes made to the original version. You should upload this as a separate file labeled 'Revised Manuscript with Track Changes'.An unmarked version of your revised paper without tracked changes. You should upload this as a separate file labeled 'Manuscript'.

If applicable, we recommend that you deposit your laboratory protocols in protocols.io to enhance the reproducibility of your results. Protocols.io assigns your protocol its own identifier (DOI) so that it can be cited independently in the future. For instructions see: https://journals.plos.org/plosone/s/submission-guidelines#loc-laboratory-protocols. Additionally, PLOS ONE offers an option for publishing peer-reviewed Lab Protocol articles, which describe protocols hosted on protocols.io. Read more information on sharing protocols at https://plos.org/protocols?utm_medium=editorial-emailutm_source=authorlettersutm_campaign=protocols.

We look forward to receiving your revised manuscript.

Kind regards,

Elsayed Abdelkreem, MD, PhD

Academic Editor

PLOS ONE

Journal Requirements:

Reviewers' comments:

Reviewer's Responses to Questions

**Comments to the Author**

1. If the authors have adequately addressed your comments raised in a previous round of review and you feel that this manuscript is now acceptable for publication, you may indicate that here to bypass the “Comments to the Author” section, enter your conflict of interest statement in the “Confidential to Editor” section, and submit your "Accept" recommendation.

Reviewer #1: All comments have been addressed

2. Is the manuscript technically sound, and do the data support the conclusions?

Reviewer #1: Yes

3. Has the statistical analysis been performed appropriately and rigorously? 

Reviewer #1: Yes

4. Have the authors made all data underlying the findings in their manuscript fully available?

Reviewer #1: No

5. Is the manuscript presented in an intelligible fashion and written in standard English?

Reviewer #1: Yes

6. Review Comments to the Author

Reviewer #1: (No Response)

7. PLOS authors have the option to publish the peer review history of their article (what does this mean?). If published, this will include your full peer review and any attached files.

Reviewer #1: **Yes: **Yitagesu Sintayehu

---

## [Author Response · Author response to Decision Letter 1]

11 Dec 2021

Thank you for considering our manuscript for publication in PLOS One. We have addressed the concerns raised by the reviewers below.

1) In the conclusion: “The low level of respondents’ awareness, practice, and facility readiness demonstrates the current weak state of secondary health systems in Kano”. Conclusion has to be based on study findings. Since this study didn’t investigate the “state of secondary health systems in Kano”, authors are encouraged to revise this sentence (reducing its strength/certainty) (e.g., replacing demonstrates by suggests or indicates).

Response:The conclusion has been revised to reflect the study findings and the word demonstrates has been changed to suggest and indicates in the abstract conclusion and main conclusion respectively.

2) In the abstract: “Respondents’ designation predicted good knowledge (aOR=0.08, 95% CI: 0.01–0.69; p= 0.01), but not practice”. Authors should better define/specify “designation” (e.g., physicians versus nurses/midwives).

Response:The designation has been better specified by including physicians versus nurses/midwives to describe it.

---

## [Editor Report · Decision Letter 2]

17 Dec 2021

PONE-D-21-11467R2Facility and care provider emergency preparedness for neonatal resuscitation in Kano, Nigeria.PLOS ONE

Dear Dr. Usman,

Thank you for submitting your manuscript to PLOS ONE. After careful consideration, we feel that it has merit but does not fully meet PLOS ONE’s publication criteria as it currently stands. Therefore, we invite you to submit a revised version of the manuscript that addresses the points raised during the review process. Specifically, in the conclusion/results "Respondents’ designation predicted good knowledge (aOR=0.08, 95% CI: 0.01–0.69; p= 0.01). In order to provide the readers with meaningful information, authors should specify "designation" (e.g.,physicians versus nurses/midwives).

We look forward to receiving your revised manuscript.

Kind regards,

Elsayed Abdelkreem, MD, PhD

Academic Editor

PLOS ONE
---

## [Author Response · Author response to Decision Letter 2]

20 Dec 2021

Specifically, in the conclusion/results "Respondents’ designation predicted good knowledge (aOR=0.08, 95% CI: 0.01–0.69; p= 0.01). In order to provide the readers with meaningful information, authors should specify "designation" (e.g.,physicians versus nurses/midwives).

Response:The word designation has been changed to profession for clarity and further specified by including physicians versus nurses/midwives to describe it in the abstract, methods, results and conclusion.

---

## [Editor Report · Decision Letter 3]

26 Dec 2021

Facility and care provider emergency preparedness for neonatal resuscitation in Kano, Nigeria.

PONE-D-21-11467R3

Dear Dr. Usman,

We’re pleased to inform you that your manuscript has been judged scientifically suitable for publication and will be formally accepted for publication once it meets all outstanding technical requirements.

Kind regards,

Elsayed Abdelkreem, MD, PhD

Academic Editor

PLOS ONE
---

## [Editor Report · Acceptance letter]

30 Dec 2021

PONE-D-21-11467R3 

Facility and care provider emergency preparedness for neonatal resuscitation in Kano, Nigeria. 

Dear Dr. Usman:

I'm pleased to inform you that your manuscript has been deemed suitable for publication in PLOS ONE. Congratulations! Your manuscript is now with our production department. 

Kind regards, 

on behalf of

Dr. Elsayed Abdelkreem 

Academic Editor

PLOS ONE